# Tabular Data: Deep Learning is Not All You Need

**Ravid Shwartz-Ziv**                                             RAVID.ZIV@INTEL.COM
*IT AI Group, Intel*

**Amitai Armon**                                                 AMITAI.ARMON@INTEL.COM
*IT AI Group, Intel*

## Abstract

A key element of AutoML systems is setting the types of models that will be used for each type of task. For classification and regression problems with tabular data, the use of tree ensemble models (like XGBoost) is usually recommended. However, several deep learning models for tabular data have recently been proposed, claiming to outperform XGBoost for some use-cases. In this paper, we explore whether these deep models should be a recommended option for tabular data, by rigorously comparing the new deep models to XGBoost on a variety of datasets. In addition to systematically comparing their accuracy, we consider the tuning and computation they require. Our study shows that XGBoost outperforms these deep models across the datasets, including datasets used in the papers that proposed the deep models. We also demonstrate that XGBoost requires much less tuning. On the positive side, we show that an ensemble of the deep models and XGBoost performs better on these datasets than XGBoost alone.

## 1. Introduction

Deep neural networks have demonstrated great success across various domains, including images, audio, and text (Devlin et al., 2018; He et al., 2016; Oord et al., 2016). Several canonical architectures exist for these domains that encode raw data efficiently into meaningful representations. These canonical architectures usually perform highly in real-world applications.

Tabular data, which consists of a set of samples (rows) with the same set of features (columns), is the most common data type in real-world applications. Many challenges arise when applying deep neural networks to tabular data, including lack of locality, data sparsity (missing values), mixed feature types (numeric, ordinal, categorical), and lack of prior knowledge of the dataset structure (unlike with text or images). Although the 'no free lunch' principle (Wolpert and Macready, 1997) always applies, tree-ensemble algorithms, such as XGBoost, are currently the recommended option for real-life tabular data problems (Chen and Guestrin, 2016; Friedman, 2001; Prokhorenkova et al., 2017).

However, recently several attempts have been made to use deep networks with tabular data (Arik and Pfister, 2019; Abutbul et al., 2020; Popov et al., 2019), some of which were claimed to outperform XGBoost. Papers in this field typically use different datasets because there is no standard benchmark. This makes comparing the models challenging, especially since some models lack open-source implementations. Furthermore, other papers that have attempted to compare these models did not optimize all the models equivalently.

The main purpose of this study is to explore whether any of the proposed deep models should indeed be a recommended choice for tabular dataset problems. There are two parts

to this question: (1) Are the models more accurate, especially for datasets that did not appear in the paper that proposed them? (2) How long do training and hyperparameter search take in comparison to other models?

We analyze the deep models proposed in four recent papers across eleven datasets, nine of which were used in these papers, to answer these questions. We show that in most cases, each model performs best on the datasets used in its respective paper but significantly worse on other datasets. Moreover, our study shows that XGBoost (Chen and Guestrin, 2016) usually outperforms the deep models on these datasets. Furthermore, we demonstrate that the hyperparameter search process was much shorter for XGBoost. On the other hand, we examine the performance of an ensemble of the deep models combined with XGBoost, and show that this ensemble achieves the best results. It also performs better than an ensemble of deep models without XGBoost, or an ensemble of classical models.

Of course any selection of tabular datasets cannot represent the full diversity of this type of data, and the 'no free lunch' principle means that no model is always better or worse than any other model. Still, our systematic study demonstrates that deep learning is currently not all we need for tabular data, despite the recent significant progress.

## 2. Deep Neural Models for Tabular Data

Among the recently proposed deep models for learning from tabular data, we examine the following: TabNet (Arik and Pfister, 2019), NODE (Popov et al., 2019), DNF-Net (Abutbul et al., 2020) and 1D-CNN (Baosenguo, 2021). To keep the paper self-contained, we briefly describe the key ideas of each of these models.

**TabNet -** TabNet is a deep learning end-to-end model that performed well across several datasets (Arik and Pfister, 2019). In its encoder, sequential decision steps encode features using sparse learned masks and select relevant features using the mask (with attention) for each row. Using sparsemax layers, the encoder forces the selection of a small set of features. The advantage of learning masks is that features need not be all-or-nothing. Rather than using a hard threshold on a feature, a learnable mask can make a soft decision, thus providing a relaxation of classical (non-differentiable) feature selection methods.

**Neural Oblivious Decision Ensembles (NODE) -** The NODE network (Popov et al., 2019) contains equal-depth oblivious decision trees (ODTs), which are differentiable so that error gradients can backpropagate through them. ODTs split data along the features and compare each with a learnable threshold. Only one feature is chosen at each level, resulting in a balanced ODT. The complete model provides an ensemble of differentiable trees.

**DNF-Net -** The idea behind DNF-Net (Abutbul et al., 2020) is to simulate disjunctive normal formulas (DNF) in DNNs. The authors proposed replacing the hard Boolean formulas with soft, differentiable versions of them. A key feature of this model is the disjunctive normal neural form (DNNF) block, which contains (1) a fully connected layer; (2) a DNNF layer formed by a soft version of binary conjunctions over literals. The complete model is an ensemble of these DNNFs.

**1D-CNN -** Recently, 1D-CNN achieved the best single model performance in a Kaggle competition with tabular data (Baosenguo, 2021). The model is based on the idea that CNN structure performs well in feature extraction, but it is rarely used in tabular data

because the feature ordering has no locality characteristics. In this model, an FC layer is used to create a larger set of features with locality characteristics, and it is followed by several 1D-Conv layers with shortcut-like connections.

**Ensemble of models -** Ensemble learning is a well-known method for improving accuracy and reducing variance through training multiple models and combining their predictions (Caruana et al., 2004). Our ensemble includes five different classifiers: TabNet, NODE, DNF-Net, 1D-CNN, and XGBoost. We construct a simple and practical ensemble using a weighted average of the single trained models predictions. The relative weights are defined simply by the normalized validation loss of each model. Note that some of the models above have some form of ensemble built into their design. However, these are ensembles of the same basic models with different parameters, not of different types of models.

## 3. Comparing the Models

We investigate whether the proposed deep models have advantages when used in various tabular datasets. For real-world applications, models must (1) perform accurately, (2) be trained and make inferences efficiently, and (3) have a short optimization time (fast hyper-parameter tuning). We first evaluate the accuracy of the deep models, XGBoost and ensembles on various datasets. Next, we analyze the different components of the ensemble. We investigate how to select models for the ensemble and test whether deep models are essential for producing good results or combining 'classical' models (XGBoost, SVM (Cortes and Vapnik, 1995) and CatBoost (Dorogush et al., 2018)) is sufficient. In addition, we explore the tradeoff between accuracy and computational resource requirements. Finally, we compare the hyperparameter search process of the different models and demonstrate that XGBoost outperforms the deep models.

### 3.1 Experimental Setup

#### 3.1.1 Datasets

As mentioned above, we investigate four deep learning models. We use nine datasets from the papers on TabNet, DNF-Net, and NODE, drawing three datasets from each paper. We additionally use two Kaggle datasets not used in any of these papers. 1D-CNN was proposed in a Kaggle competition recently for use on one specific dataset, which we do not explore. The datasets we use are Forest Cover Type, Higgs Boson, Year Prediction, Rossmann Store Sales, Gas Concentrations, Eye Movements, Gesture Phase, MSLR, Epsilon, Shrutime and Blastchar. For dataset details, see Appendix B.

#### 3.1.2 The Optimization Process

To find the hyper-parameters, we used HyperOpt (Bergstra et al., 2015), which uses Bayesian optimization. The hyperparameter search was run for 1000 steps on each dataset by optimizing the results on a validation set. The initial hyperparameters were taken from the original paper. The hyperparameter search space for each model is provided in Appendix C. We split the datasets to training, validation and test sets in the same way as in the original papers that used them. When the split was reported to be random, we performed three repetitions of the random partition (as done in the original paper), and we report

their mean (for the standard error of the mean see Appendix A). Otherwise, we used three random seed initializations in the same partition, and we report their average. For the classification datasets, we minimize cross-entropy loss and report the classification error. For the regression datasets, we minimize and report mean squared error. We use the term 'original model' to refer to the model used on a given dataset in the paper that presented the respective model. The 'unseen datasets' for each model are those not mentioned in the paper that published the respective model. Note that a model's unseen dataset is not a dataset it was not trained on, but to a dataset that did not appear in its original paper.

## 3.2 Results

Do the deep models generalize well to other datasets?

We first explore whether the deep models perform well when trained on datasets that were not included in their original paper, and compare them to XGBoost. Table 1 presents the accuracy measures of each model for each dataset (lower indicates greater accuracy). The first three columns correspond to datasets from the TabNet paper, the following three to the DNF-Net paper, and the next three to the NODE paper. The last two columns correspond to datasets that did not appear in any of these papers.

We make several observations regarding these results:

- In most cases, the models perform worse on unseen datasets than do the datasets' original models.

- The XGBoost model generally outperformed the deep models.

- No deep model consistently outperformed the others. The 1D-CNN model performance may seem to perform better, since all the datasets were new for it.

- The ensemble of deep models and XGBoost outperforms the other models in most cases.

Furthermore, we calculated for each dataset the relative performance of each model compared to the best model for that dataset. We averaged this per model on all its unseen datasets (geometric mean). The ensemble of all the models was the best model with 2.32% average relative increase, XGBoost was the second best with 3.4%, 1D-CNN had 7.5%, TabNet had 10.5%, DNF-Net had 11.8% and NODE had 14.2% (see Tables 2 and 3 in the appendix for full results).

These results are somewhat surprising. When we train on datasets other than those in their original papers, the deep models perform worse than XGBoost. Compared to XGBoost and the full ensemble, the single deep models performance is much more sensitive to the specific dataset. There may be several reasons for the deep models to perform worse when they are trained on previously unseen datasets. The first possibility is **selection bias**. Each paper may have naturally demonstrated the model's performance on datasets with which the model worked well. The second possibility is differences in the **optimization of hyperparameters**. Each paper may have set the model's hyperparameters based on a more extensive hyperparameter search on the datasets presented in that paper, resulting in better performance. Our results for each model on its original datasets matched those presented

| Name | Rossman | CoverType | Higgs | Gas | Eye | Gesture | YearPrediction | MSLR | Epsilon | Shrutime | Blastchar |
|------|---------|-----------|-------|-----|-----|---------|----------------|------|---------|----------|-----------|
| XGBoost | 490.18 | 3.13 | 21.62 | 2.18 | **56.07** | 80.64 | 77.98 | 55.43 | 11.12 | 13.82 | 20.39 |
| NODE | 488.59 | 4.15 | 21.19 | 2.17 | 68.35 | 92.12 | 76.39 | 55.72 | **10.39** | 14.61 | 21.40 |
| DNF-Net | 503.83 | 3.96 | 23.68 | **1.44** | 68.38 | 86.98 | 81.21 | 56.83 | 12.23 | 16.80 | 27.91 |
| TabNet | **485.12** | 3.01 | **21.14** | 1.92 | 67.13 | 96.42 | 83.19 | 56.04 | 11.92 | 14.94 | 23.72 |
| 1D-CNN | 493.81 | 3.51 | 22.33 | 1.79 | 67.90 | 97.89 | 78.94 | 55.97 | 11.08 | 15.31 | 24.68 |
| Simple Ensemble | 488.57 | 3.19 | 22.46 | 2.36 | 58.72 | 89.45 | 78.01 | 55.46 | 11.07 | 13.61 | 21.18 |
| Deep Ensemble w/o XGBoost | 489.94 | 3.52 | 22.41 | 1.98 | 69.28 | 93.50 | 78.99 | 55.59 | 10.95 | 14.69 | 24.25 |
| Deep Ensemble w XGBoost | 485.33 | **2.99** | 22.34 | 1.69 | 59.43 | **78.93** | **76.19** | **55.38** | 11.18 | **13.10** | **20.18** |

TabNet      DNF-Net      NODE      New datasets

Table 1: **Test results on tabular datasets.** The table presents for each model the MSE for the YearPrediction and the Rossman datasets and the logloss (with 100X factor) for the other datasets. The values are the averages of three training runs (lower is better). The papers which used these datasets are indicated below the table.

in its respective paper, thus excluding implementation issues as the possible reason for our observations.

### Do we need both XGBoost and deep networks?

In the previous subsection we saw that the ensemble of XGBoost and deep models performed best across the datasets. It is therefore interesting to examine which component of our ensemble is mandatory. One question is whether XGBoost needs to be combined with the deep models, or would a simpler ensemble of non-deep models perform similarly. To explore this, we trained an ensemble of XGBoost and other non-deep models: SVM (Cortes and Vapnik, 1995) and CatBoost(Dorogush et al., 2018). Table 1 shows that the ensemble of classical models performed much worse than the ensemble of deep networks and XGBoost. Additionally, the table shows that the ensemble of deep models alone (without XGBoost) did not provide good results. This indicates that combining both the deep models and XGBoost provides an advantage for these datasets.

### Subset of models

We observed that the ensemble improved accuracy, but the use of multiple models also requires additional computation. When real-world applications are considered, computational constraints may affect the eventual performance. We therefore considered using subsets of the models within the ensemble, to see the tradeoff between accuracy and computation.

There are several ways to choose a subset from an ensemble of models: (1) based on the **validation loss**, choosing models with low validation loss first, (2) based on the models' **uncertainty for each example**, choosing the highest confidence models (by some uncertainty measure) for each example, and (3) based on a **random order**.

In Figure 1 these methods of selecting models are compared for an example of an unseen dataset (Shrutime). The best selection approach was averaging the predictions based on the models' validation loss. Only three models were needed to achieve almost optimal performance this way. Choosing the models randomly provided the worst choice according to our comparison.

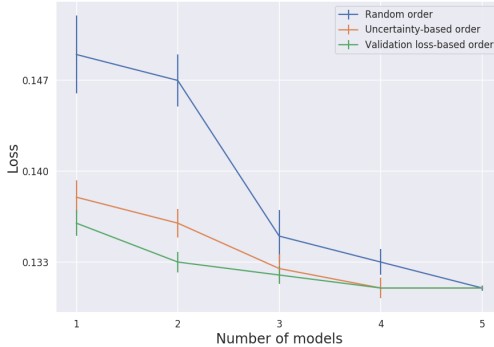

Figure 1: **The impact of selecting a subset of models in the ensemble**.

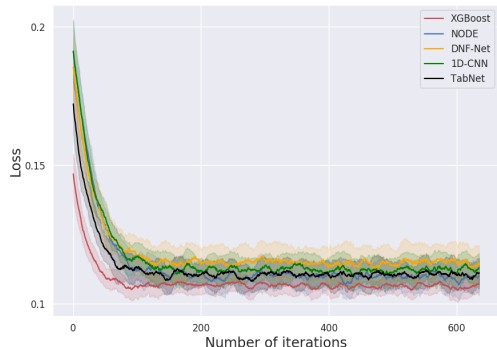

Figure 2: **The Hyper-parameters optimization process for different models**.

How difficult is the optimization?

In real-life applications, we often have a limited amount of time to optimize our model for use on a new dataset. This is a significant consideration in AutoML systems, which run multiple models on many datasets. We are therefore interested in the total number of iterations it takes to optimize a model. Figure 2 shows the model's performance (mean and standard error of the mean) as a function of the number of iterations of the hyper-parameter optimization process for the Shrutime dataset. We observe that XGBoost outperformed the deep models, converging to good performance more quickly (in fewer iterations, which were also shorter in terms of runtime). These results may be affected by several factors: (1) We used a **Bayesian hyperparameter optimization process**, and the results may differ for other optimization processes; (2) **the initial hyperparameters** of XGBoost may be more robust because it had previously been optimized over many datasets. Perhaps we could find some hyperparameters that would also work well for the deep models for different datasets; and (3) the XGBoost model may have some **inherent characteristics** that make it more robust and easier to optimize. It may be interesting to further investigate this behavior.

## 4. Summary

In this paper we investigated the accuracy of recently proposed deep models for tabular datasets. According to our analysis, these deep models were weaker on datasets that did not appear in their original papers, and they were weaker than XGBoost, the baseline model. We proposed using an ensemble of these deep models with XGBoost, which performed better on these datasets than any individual model and the 'non-deep' classical ensemble. We also explored some examples for the possible tradeoffs between accuracy, inference computational cost, and hyperparameter optimization time, which are important for real-world applications, especially for AutoML. In conclusion, while significant progress has been made using deep models for tabular data, they still do not outperform XGBoost, and further research is apparently needed in this field. Our somewhat improved ensemble results provide another potential avenue for further research.

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

| Name | Rossman | CoverType | Higgs | Gas | Eye | Gesture | YearPrediction | MSLR | Epsilon | Shrutime | Blastchar |
|---|---|---|---|---|---|---|---|---|---|---|---|
| XGBoost | 490.18 ± 2.19 | 3.13 ± 0.09 | 21.62 ± 0.33 | 2.18 ± 0.20 | **56.07**±0.65 | 80.64 ± 0.80 | 77.98 ± 0.11 | 55.43±2e-2 | 11.12±3e-2 | 13.82 ± 0.19 | 20.39 ± 0.21 |
| NODE | 488.59 ± 1.24 | 4.15 ± 0.13 | 21.19 ± 0.69 | 2.17 ± 0.18 | 68.35 ± 0.66 | 92.12 ± 0.82 | 76.39 ±0.13 | 55.72±3e-2 | **10.39**±1e-2 | 14.61 ± 0.10 | 21.40 ± 0.25 |
| DNF-Net | 503.83 ± 1.41 | 3.96 ± 0.11 | 23.68 ± 0.83 | **1.44** ±0.09 | 68.38 ± 0.65 | 86.98 ± 0.74 | 81.21 ± 0.18 | 56.83±3e-2 | 12.23±4e-2 | 16.8 ± 0.09 | 27.91 ± 0.17 |
| TabNet | **485.12**±1.93 | 3.01 ± 0.08 | **21.14**±0.20 | 1.92 ± 0.14 | 67.13 ± 0.69 | 96.42 ± 0.87 | 83.19 ± 0.19 | 56.04±1e-2 | 11.92±3e-2 | 14.94±, 0.13 | 23.72 ± 0.19 |
| 1D-CNN | 493.81 ± 2.23 | 3.51 ± 0.13 | 22.33 ± 0.73 | 1.79 ± 0.19 | 67.9 ± 0.64 | 97.89 ± 0.82 | 78.94 ± 0.14 | 55.97±4e-2 | 11.08±6e-2 | 15.31 ± 0.16 | 24.68 ± 0.22 |
| Simple Ensemble | 488.57 ± 2.14 | 3.19 ± 0.18 | 22.46 ± 0.38 | 2.36 ± 0.18 | 58.72 ± 0.67 | 89.45 ± 0.89 | 78.01 ± 0.17 | 55.46±4e-2 | 11.07±4e-2 | 13.61±, 0.14 | 21.18 ± 0.17 |
| Deep Ensemble w/o XGBoost | 489.94 ± 2.09 | 3.52 ± 0.10 | 22.41 ± 0.54 | 1.98 ± 0.13 | 69.28 ± 0.62 | 93.50 ± 0.75 | 78.99 ± 0.11 | 55.59±3e-2 | 10.95±1e-2 | 14.69 ± 0.11 | 24.25 ± 0.22 |
| Deep Ensemble w XGBoost | 485.33 ± 1.29 | **2.99** ±0.08 | 22.34 ± 0.81 | 1.69 ± 0.10 | 59.43 ± 0.60 | **78.93** ±0.73 | **76.19** ±0.21 | **55.38**±1e-2 | 11.18±1e-2 | **13.10**±0.15 | **20.18**±0.16 |

TabNet        DNF-Net        NODE        New datasets

Table 2: Full test results on the tabular datasets. The table presents for each model the MSE for the YearPrediction and the Rossman datasets and the logloss (with 100X factor) for the other datasets. The values are the averages of three training runs (lower is better), along with the standard error of the mean (SEM).

Kaggle. Rossmann store sales, 2019b. URL `https://www.kaggle.com/c/rossmann-store-sales`.

Rory Mitchell, Andrey Adinets, Thejaswi Rao, and Eibe Frank. Xgboost: Scalable gpu accelerated learning. *arXiv preprint arXiv:1806.11248*, 2018.

Aaron van den Oord, Sander Dieleman, Heiga Zen, Karen Simonyan, Oriol Vinyals, Alex Graves, Nal Kalchbrenner, Andrew Senior, and Koray Kavukcuoglu. Wavenet: A generative model for raw audio. *arXiv preprint arXiv:1609.03499*, 2016.

Pascal. Pascal large scale learning challenge, 2008. URL `https://www.csie.ntu.edu.tw/~cjlin/libsvmtools/datasets/binary.html`.

Sergei Popov, Stanislav Morozov, and Artem Babenko. Neural oblivious decision ensembles for deep learning on tabular data. *arXiv preprint arXiv:1909.06312*, 2019.

Liudmila Prokhorenkova, Gleb Gusev, Aleksandr Vorobev, Anna Veronika Dorogush, and Andrey Gulin. Catboost: unbiased boosting with categorical features. *arXiv preprint arXiv:1706.09516*, 2017.

Tao Qin and Tie-Yan Liu. Introducing LETOR 4.0 datasets. *CoRR*, abs/1306.2597, 2013. URL `http://arxiv.org/abs/1306.2597`.

Joaquin Vanschoren, Jan N Van Rijn, Bernd Bischl, and Luis Torgo. Openml: networked science in machine learning. *ACM SIGKDD Explorations Newsletter*, 15(2):49–60, 2014.

David H Wolpert and William G Macready. No free lunch theorems for optimization. *IEEE transactions on evolutionary computation*, 1(1):67–82, 1997.

## Appendix A. Full Results

In Table 2 we show the mean and the standard error of the mean (SEM) for each dataset and for each model. In Table 3 we present the average relative performance deterioration of each model on its unseen datasets (geometric mean, lower is better).

| Name | Average Relative Performance (%) |
|---|---|
| XGBoost | 3.34 |
| NODE | 14.21 |
| DNF-Net | 11.96 |
| TabNet | 10.51 |
| 1D-CNN | 7.56 |
| Simple Ensemble | 3.15 |
| Deep Ensemble w/o XGBoost | 6.91 |
| Deep Ensemble w XGBoost | **2.32** |

Table 3: Average relative performance deterioration for each model on its unseen datasets (lower is better)

| Dataset | Features | Classes | Samples | Source | Paper | Link |
|---|---|---|---|---|---|---|
| Gesture Phase | 32 | 5 | 9.8k | OpenML | DNF-Net | openml.org/d/4538 |
| Gas Concentrations | 129 | 6 | 13.9k | OpenML | DNF-Net | openml.org/d/1477 |
| Eye Movements | 26 | 3 | 10.9k | OpenML | DNF-Net | openml.org/d/1044 |
| Epsilon | 2000 | 2 | 500k | PASCAL Challenge 2008 | NODE | https://www.csie.ntu.edu.tw/ cjlin/libsvmtools/datasets/binary.html |
| YearPrediction | 90 | 1 | 515k | Million Song Dataset | NODE | https://archive.ics.uci.edu/ml/datasets/yearpredictionmsd |
| Microsoft (MSLR) | 136 | 5 | 964k | MSLR-WEB10K | NODE | https://www.microsoft.com/en-us/research/project/mslr/ |
| Rossmann Store Sales | 10 | 1 | 1018K | Kaggle | TabNet | https://www.kaggle.com/c/rossmann-store-sales |
| Forest Cover Type | 54 | 7 | 580k | Kaggle | TabNet | https://www.kaggle.com/c/forest-cover-type-prediction |
| Higgs Boson | 30 | 2 | 800k | Kaggle | TabNet | https://www.kaggle.com/c/higgs-boson |
| Shrutime | 11 | 2 | 10k | Kaggle | New dataset | https://www.kaggle.com/shrutimechlearn/churn-modelling |
| Blastchar | 20 | 2 | 7k | Kaggle | New dataset | https://www.kaggle.com/blastchar/telco-customer-churn |

Table 4: Description of the tabular datasets

## Appendix B. Tabular Data-sets Description

In our experiments we use datasets that differ in their characteristics, such as the number of features, the number of classes and the number of samples (see Table 4). For each dataset, we followed the prepossessing and training procedures described in the original paper. The datasets we use are Forest Cover Type, Higgs Boson and Year Prediction datasets (Dua and Graff, 2017), Rossmann Store Sales (Kaggle, 2019b), Gas Concentrations, Eye Movements and Gesture Phase (Vanschoren et al., 2014), MSLR (Qin and Liu, 2013), Epsilon (Pascal, 2008), Shrutime (Kaggle, 2019a), and Blastchar (IBM, 2019).

## Appendix C. Optimization of hyperparameters

In order to tune the hyperparameters, we split the datasets to training, validation and test sets in the same way as in the original papers that used them. Specifically, we performed a random stratified split of the full training data into train set (80%) and validation set (20%) for the Epsilon, YearPrediction, MSLR, Shrutime and Blastchar datasets. For Eye, Gesture and Year datasets, we split the full data to validation set (10%), test set (20%) and train set (70%). For Forest Cover Type, we used train/val/test split provided by the dataset authors (Mitchell et al., 2018). For the Rossmann dataset we used the same preprocessing and data split as (Prokhorenkova et al., 2017) – data from 2014 was used for training and validation, whereas 2015 was used for testing. We split $100k$ samples for validation from the training dataset, and after the optimization of the hyperparameters we retrained with the entire training dataset. For the Higgs dataset, we split $500k$ samples for validation from the training dataset, and after the optimization of the hyperparameters we retrained with the entire training dataset. We used the Hyperopt library to optimize the models. For the

final configuration we selected the set of hyperparameters corresponding to the smallest loss on the validation set. For all the models, early stopping is applied using the validation set.

## C.1 CATBOOST

The list of hyperparameters and their search spaces for Catboost:

- Learning rate: Log-Uniform distribution $[e^{-5}, 1]$

- Random strength: Discrete uniform distribution $[1, 20]$

- Max size: Discrete uniform distribution $[0, 25]$

- L2 leaf regularization: Log-Uniform distribution $[1, 10]$

- Bagging temperature: Uniform distribution $[0, 1]$

- Leaf estimation iterations: Discrete uniform distribution $[1, 20]$

## C.2 XGBoost

The list of hyperparameters and their search spaces for XGBoost:

- Number of estimators: Uniform distribution $[100, 4000]$

- Eta: Log-Uniform distribution $[e^{-7}, 1]$

- Max depth: Discrete uniform distribution $[1, 10]$

- Subsample: Uniform distribution $[0.2, 1]$

- Colsample bytree: Uniform distribution $[0.2, 1]$

- Colsample bylevel: Uniform distribution $[0.2, 1]$

- Min child weight: Log-Uniform distribution $[e^{-16}, e^5]$

- Alpha: Uniform choice $\{0, \text{Log-Uniform distribution } [e^{-16}, e^2]\}$

- Lambda: Uniform choice $\{0, \text{Log-Uniform distribution } [e^{-16}, e^2]\}$

- Gamma: Uniform choice $\{0, \text{Log-Uniform distribution } [e^{-16}, e^2]\}$

## C.3 NODE

The list of hyperparameters and their search spaces for NODE:

- Learning rate: Log-Uniform distribution $[e^{-5}, 1]$

- Num layers: Discrete uniform distribution $[1, 10]$

- Total tree count: $\{256, 512, 1024, 2048\}$

- Tree depth: Discrete uniform distribution $[4, 9]$

- Tree output dim: Discrete uniform distribution $[1, 5]$

- Learning rate - Log-Uniform distribution $[e^{-4}, 0.5]$

- Batch size - Uniform choice $\{512, 1024, 2048, 4096, 8192\}$

### C.4 TabNet

The list of hyperparameters and their search spaces for TabNet:

- Learning rate: Log-Uniform distribution $[e^{-5}, 1]$

- feature dim: Discrete uniform distribution $[20, 60]$

- output dim: Discrete uniform distribution $[20, 60]$

- n steps: Discrete uniform distribution $[1, 8]$

- bn epsilon: Uniform distribution $[e^{-5}, e^{-1}]$

- relaxation factor: Uniform distribution $[0.3, 2]$

- Batch size - Uniform choice $\{512, 1024, 2048, 4096, 8192\}$

### C.5 DNF-Net

The list of hyperparameters and their search spaces for DNF-Net:

- n. formulas: Discrete uniform distribution $[256, 2048]$

- Feature selection beta: Discrete uniform distribution $[1e^{-2}, 2]$

- Learning rate - Log-Uniform distribution $[e^{-4}, 0.5]$

- Batch size - Uniform choice $\{512, 1024, 2048, 4096, 8192\}$

### C.6 1D-CNN

The list of hyperparameters and their search spaces for 1D-CNN:

- hidden layer sizes: Discrete uniform distribution $[100, 4000]$

- Number of layers:Discrete uniform distribution $[1, 6]$

- Learning rate - Log-Uniform distribution $[e^{-4}, 0.5]$

- Batch size - Uniform choice $\{512, 1024, 2048, 4096, 8192\}$

