# OpenReview forum: "Tabular Data: Deep Learning is Not All You Need"
_ICML.cc/2021/Workshop/AutoML — AutoML@ICML2021 Poster_

### Official Review · Reviewer_DHF8 · 2021-06-13
**Bold claims without strong basis**

**Rating:** 3
**Confidence:** 5

**Review:**

Overall, the paper is on a very important and timely topic, and I appreciate all the work the authors have put into it.

My major concern is that, for such an empirical paper with bold takeaways as the results of experiments, the assumptions and the settings should be solid, and not questionable. But it is not the case for this paper. The hyperparameter search spaces seem arbitrary. For example, the learning rate is not tuned for TabNet, or very small unit sizes are used. I strongly urge the authors to carefully look at all the papers of the comparison models, see the hyperparameters that they have used in their experiments, and adapt them here as well. If such assumptions will be questionable, takeaways like "Our study shows that XGBoost outperforms these deep models across the datasets" are not acceptable, and indeed problematic for the community.

In addition, I would like to note that the tabular datasets are very diverse, and conclusions from 11 datasets may not mean anything. For example, if I cut the dataset sizes by half, or if I double them, how would this ranking change given that data efficiency of different models are very different? Or if I have a dataset in the form of y=x1+x2, wouldn't linear regression outperform everything? Rather than focusing on these bold claims, I suggest being more humble and simply reporting concrete experimental observations and hypotheses on them.

The paper has positive aspects like studying the hyperparameter tuning need, or ensembling of different types of tabular learning models. These are highly important in practice. However, because of the reasons mentioned above, I strongly support rejection. I indeed believe propagation of such papers (with bold claims from limited experimental evidence) is unfortunate for the future of machine learning. My advice for the authors is properly performing the experiments from scratch, and rewriting the paper to avoid such bold claims.

---

### Official Review · Reviewer_URYQ · 2021-06-14
**Paper Review**

**Rating:** 5
**Confidence:** 5

**Review:**

This paper provides a benchmark comparing several recently introduced tabular deep learning models along with XGBoost to determine if deep learning models can compare favorably to more traditional tree-ensemble models. The paper additional considers the impact of ensembling different models, including an ensemble of traditional models, deep models, and a combined ensemble of deep models with XGBoost. The paper concludes that deep learning models do not consistently match or outperform XGBoost, but that they do provide value via ensembling with XGBoost, outperforming a traditional ensemble and XGBoost alone.

I do tend to agree with the overall conclusion of the paper that the primary benefit of deep models in tabular problems is via ensembling with traditional models. In particular, it is valuable to see that the neural network models appear to struggle to generalize to datasets outside of their original benchmarks. However, the paper does seem to have significant issues with clarity in its evaluation process which in my opinion leads to some confusing results that could have been avoided by using more new datasets as opposed to primarily re-using datasets from the various neural network papers. The paper also seems to be missing important related works in AutoML that have demonstrated significant improvement by ensembling deep learning models with traditional models.

Detailed Comments:

"The main purpose of this study is to explore whether any of the proposed deep models should indeed be a recommended choice for tabular dataset problems. There are two parts to this question: (1) Are the models more accurate, especially for datasets that did not appear in the paper that proposed them? (2) How long do training and hyperparameter search take in comparison to other models?"

- Part 2 does not seem to be answered in this paper, as no definitive measure of training time nor hyperparameter search time is provided.
- This does not mention whether they can provide value via ensembling, which is a major part of the paper.

"Furthermore, we calculated for each dataset the relative performance of each model compared to the best model for that dataset. We averaged this per model on all its unseen datasets (geometric mean)."

- This seems to have a couple issues:

1. If the range of error in dataset 1 is between 0.10 and 0.01, it will inherently have a much higher relative performance impact when averaged than dataset 2 with range of error between 0.20 and 0.10. Because of this, datasets like Rossman, Higgs, YearPrediction, and Microsoft have hardly any impact on the results compared to datasets like CoverType and Gas.

2. Because a different subset of datasets is being averaged for each model, the comparison is shaky, particularly because of the issue stated prior. I believe this is the primary reason NODE is marked as coming in last in the comparison, as its subset of datasets did not include YearPrediction and Microsoft, where even the worst model only had a 8% (YearPrediction) and 2.5% (Microsoft) relative increase.

It would benefit the paper to have more new datasets and potentially remove the non-new datasets entirely from the final averaged relative loss increase to avoid misleading conclusions.

Important related work is missing in regards to existing methods that ensemble deep learning models with traditional models, particularly in the AutoML domain [1,2].

Given that weighted model ensembling was investigated significantly in this work, common methods used in many practical AutoML systems should be referenced and perhaps benchmarked [3].

A better metric than number of hyperparameter tuning iterations would be the overall runtime of optimization for each model. This is particularly important when training models with very different architectures and thus different training costs. Given up to 1000 hyperparameter tuning iterations were performed, it would also be useful to see the overall optimization runtime for each model for each dataset.

Regarding hyperparameter search spaces, I was unable to find any mention of total training epochs/iterations for individual models as a hyperparameter searched or whether an early stopping method was used. Total training epochs/iterations is often the most important hyperparameter in machine learning models, especially for models such as XGBoost. While typically they are best optimized via early stopping, the paper makes no mention of the strategy used, and the default of 100 trees is often highly suboptimal. If these hyperparameters were overlooked it would heavily impact the results.

References:

[1] Erickson, Nick et al. “AutoGluon-Tabular: Robust and Accurate AutoML for Structured Data.” ArXiv abs/2003.06505 (2020): n. pag.

[2] LeDell, E.. “H2O AutoML: Scalable Automatic Machine Learning.” (2020).

[3] Caruana, R. et al. “Ensemble selection from libraries of models.” Proceedings of the twenty-first international conference on Machine learning (2004): n. pag.

---

### Official Review · Reviewer_3RnR · 2021-06-14
**Nice experimental survey that opposes the claims of some recent papers**

**Rating:** 6
**Confidence:** 5

**Review:**

The paper presents an experimental comparison of Gradient-Boosted Decision Trees compared to three state of the art neural network architectures for tabular datasets. The authors expressed the criticism on the experimental protocols of the neural network architectures, and highlight that the published results do not generalize to new datasets. As a conclusion, the authors conduct experiments on 9 datasets to demonstrate that XGBoost is still a strong method for tabular datasets.

In my opinion, although there is no technical novelty in this paper, the message is very important. Otherwise, the community might be mislead into believing that recent deep learning publications are really state of the art on tabular data.

A point of criticism on this paper is that the choice of datasets is quite limited and no statistical significance was measured. For example you could have compared the rival methods on the OpenML AutoML benchmark of datasets https://openml.github.io/automlbenchmark/benchmark_datasets.html

---

### Decision · Program_Chairs · 2021-06-21

**Decision:**

Accept (Poster)

**Comment:**

This paper experimentally compares gradient-boosted decision trees versus neural networks for tabular data. The reviewers agree that the topic is important and that the experiments are interesting.

The main concern from the reviewers is that the breadth of the experiments do not align with the strong claims made. In particular, experiments across eleven datasets are not sufficient to make statements such as “In conclusion, while significant progress has been made using deep models for tabular data, they still do not outperform XGBoost, and further research is needed in this field.”

We recommend acceptance because the paper is relevant to the workshop and the experiments are beneficial. However, for the camera ready version, we strongly recommend that the authors revise some of the broad claims made.